# European Regulations on Camel Germplasm Movement within the European Union: A Current Framework Based on Safety

**DOI:** 10.3390/ani12172255

**Published:** 2022-08-31

**Authors:** Elena Zema, Salvatore Monti, Vito Biondi, Asim Faraz, Michela Pugliese, Gabriele Marino, Annamaria Passantino

**Affiliations:** 1Veterinary Practitioner, Via Caserta Crocevia, 89124 Reggio Calabria, Italy; 2Department of Veterinary Sciences, University of Messina, 98168 Messina, Italy; 3Department of Livestock and Poultry Production, Bahauddin Zakariya University Multan, Multan 6000, Pakistan

**Keywords:** camels, semen, oocytes, embryos, movements, European legislation

## Abstract

**Simple Summary:**

Delegated Regulation (EU) 2020/686 (hereafter Reg. 686) regulates the traceability and animal health for the movement of germinal material of camels within the EU. Given that the camel breeding industry is in a strong phase of growth, an amount of germinal material of terrestrial animals—including those belonging to the *Camelidae* family—is moved between the member states. The authors performed an analysis of Reg. 686 addressing veterinarians and breeders who want to sell high-quality germinal material or genetically improve their herds.

**Abstract:**

With the aim of developing livestock breeding, the Delegated Regulation (EU) 2020/686 (hereafter referred to as Reg. 686) has taken steps to define traceability and animal health for the movement of germ material within the European Union (EU), including that of camelid species. Despite the economic importance of the camel market and the efforts of the EU to regulate their movements, there are considerable difficulties in the collection of semen and its freezing, limiting the use of artificial insemination in this species. If, on the one hand, there is little diffusion of the camel breeding and, consequently, limited diffusion of animals and germplasm, there will probably be a significant increase over the years. To avoid the spread of emerging diseases—or even those no longer present in Europe—the entry of genetic material from non-EU countries must be strictly monitored. Camels are rarely clinically compliant, but can transfer even fatal diseases to domestic ungulate farms in the EU. Based on these considerations, we conducted a narrative review of the European regulations on this issue, focusing on aspects related to their application in camels.

## 1. Introduction

Camel breeding has always played an important role in world agriculture, and is now also becoming a source of income for breeders in the European Union (EU). It is necessary to encourage the production of animals with appropriate genetic characteristics by applying defined criteria that meet good performance while protecting animal health requirements. Disparities in these standards may create technical barriers to trade in breeding animals and their germinal products. For this reason, the European legislator issuing delegated Regulation (EU) 2020/686 (hereafter referred to as Reg. 686) has established rules regarding the approval of germinal products (e.g., semen, oocytes, and embryos)—and the traceability of these products in the EU—from certain terrestrial animals, including camelid species. Relating to the other aspects concerning entry of camelid animals into the EU and their movements between member states, Commission Regulations (EU) 2021/403 and 2021/404 should be applied.

Animal genetic resources are an essential basis for the sustainable development of the livestock sector, and offer the opportunity for animals to adapt to changing production conditions and markets.

The reproductive efficiency of camels is considered poor, resulting in low production and profitability. The main causes of their reproductive inefficiency include the late onset of puberty, seasonality, a long gestation period, high neonatal mortality, and a long interval between births. Moreover, the fact that diagnostic procedures and investigations related to infertility are not routinely implemented according to standard procedures might also be a contributing factor. Assisted reproductive techniques and related biotechnology may improve genetics, increasing birth rates and, thus, the production of this species [1].

Although the use of AI for camels has been described since the 1960s [2], it has been only recently employed for potential use on a large scale to improve genetic traits such as milk production, meat, wool, and the ability to compete in the Middle East [2,3]. However, some drawbacks have been pointed out regarding AI in camels—the reproductive behavior of the male, the type and quantity of ejaculate, the lack of knowledge of the optimal insemination time and semen dosage per insemination session, and the lack of a standard storage technique may be limiting factors [3,4]. Furthermore, the poor welfare and inappropriate management of these animals may be a critical point for the quality of semen. In fact, Fatnassi et al. (2021) [5] highlighted that male camels utilized for AI should be reared taking into consideration strategies to improve their housing conditions and safeguard their welfare, given that these factors are also linked with production and reproduction. According to some authors [6,7,8], it may be useful to increase the space allowance, provide an appropriate diet, and facilitate greater social contact with conspecifics of the same or different gender.

Despite significant progress having been made regarding AI in camels, such assisted reproductive technology is probably still far from giving the results that might be obtained with the transfer of chilled or frozen–thawed embryos, when also considering the impact of such technology in speeding up the genetic improvement [9].

The camel breeding industry is set to grow significantly [10]. Globally, although it is not possible to accurately determine the current official number of camelids [10], camels are valorized on the international market for milk and meat production and/or for export of live animals [10].

As an increasing amount of germinal material of terrestrial animals is moved between the EU member states (MSs) [11], including animals of the *Camelidae* family, we found it interesting to carry out an analysis of the Delegated Regulation (EU) 2020/686 (hereafter referred to as Reg. 686) [12] with respect to traceability and animal health for the movement of germinal material of camels within the EU.

## 2. EU Legal Framework

Since the 1980s, the EU has established rules on germline products from bovine, ovine, caprine, porcine, and equine species [13,14,15,16]. These directives have been replaced by Regulation (EU) 2016/429 [17]—the so-called “Animal Health Law (AHL)”—which lays down rules on transmissible animal diseases, registration and approval of germinal product establishments, the traceability of consignments, and animal health requirements for germinal products transported within the EU. This regulation requires germinal material to be collected, produced, processed, and stored in specialized establishments subject to specific hygiene and animal health regimes. According to Article 94, different types of establishments are approved for the protection of animal health, and for trade in animals and animal products between MSs. As “germinal material establishments”, only those in which the germinal material originates from conventional farm animals may be approved.

This regulation was later supplemented by Reg. 686 [12], which outlines the rules and requirements for the movement within the EU of germinal products of different species, including those from the *Camelidae* family. The main goal of this regulation is to prevent the spread of transmissible animal diseases within the EU through such material. Table 1 lists the articles providing rules and requirements with respect to the *Camelidae* family.

Special conditions for such storage have been established, including separate containers and storage areas for each type of germ product, biosecurity standards, traceability, and listing requirements. Particular attention is devoted to the disease status of the animals from which the germinal products originate.

Following the requirements of the AHL, Article 3 of Reg. 686 lists the subtypes of germinal material establishments that must be approved by the competent authority, and Article 4 describes the conditions that must be met for the competent authority to approve.

The subtypes eligible for approval are as follows:(i)Semen collection centers, i.e., establishments where semen from “farm animals” is collected, processed, and stored.(ii)Embryo collection groups, comprising either a group of professionals or a “facility” under the control of a veterinarian competent for embryo collection, processing, and storage operations.(iii)Embryo production groups, which are broadly similar to the “embryo collection group”, including the collection, processing, and storage of oocytes and embryos, as well as proper embryo production.(iv)Germ material processing establishments, i.e., establishments where fresh chilled or frozen semen, oocytes, or embryos are processed or stored.(v)Germinal material storage establishments, i.e., establishment, where germinal material is stored.

According to Article 4, however, the conditions to be met for the approval of the competent authority are as follows:(i)That the operator of the establishment has appointed a center or team veterinarian responsible for the established activities listed in Annex I of the same regulation.(ii)That the facilities, equipment, and operating procedures of the various subtypes of establishments meet the requirements set out in Annex I.

After ascertaining compliance with these parameters to ensure maximum traceability of germ material, the authority assigns the establishment a unique identification and approval number.

With regard specifically to camelids, on the other hand, any establishment of camelid germinal material must be registered under Article 84 of the AHL. Here, it is stated that the registration of establishments by operators who collect, produce, process, or store germinal material is mandatory. These are subsequently registered by the competent authority under Article 93.

Regulation 686 indicates that trade between MSs in camelid germinal material may only take place under the following conditions:(i)Compliance with the traceability requirements set out in Article 11;(ii)Compliance with the animal health requirements of Article 38;(iii)The presence of a health certificate accompanying the consignment of germinal material in accordance with Article 161 of Reg. 429/2016.

The information that must be included in the health certificate is contained in Annex IV of Reg. 686, and in Reg. 403/2021 there are templates. This means that the possibility of exchanging germinal material between MSs is unrelated to the presence of an actual germinal material establishment and, therefore, measures for export to other MSs must only concern donor animals (due to animal health requirements). If an establishment were to be set up in an MS exclusively for these operations, such as a semen collection center, it would have to be identified as a registered establishment and, possibly—to ensure greater protection of animal health—be a confined establishment, being approved by the competent authority.

In fact, according to the requirements for the traceability of germinal material from these animals, as outlined in Article 11 of Reg. 686, the only establishments that can be registered and approved for trade in germinal material of camelids between MSs are contained establishments, i.e., those in which the animals are kept with precise guarantees of biosecurity, and in which the germinal material is also collected, produced, processed, and stored so that it can be transferred to the contained establishments of other MSs.

However, the possible definition of a registered germplasm establishment would not directly imply the possibility of exporting germplasm to other member states.

According to the definition contained in the AHL, confined establishments include any permanent establishment located in a circumscribed geographical area, created voluntarily, and approved for movements, where the animals are:*(i)* Held or bred for participation in exhibitions, for education, for species conservation, or for research purposes.*(ii)* Confined and separated from their environment.*(iii)* Subject to health surveillance and biosecurity measures.

According to the Terrestrial Animal Health Code (OIE), a semen collection center is a “compartment” i.e., “an animal subpopulation contained in one or more establishments, separated from other susceptible populations by a common biosecurity management system, and having a specific health status for one or more infections or infestations for which the necessary surveillance, biosecurity, and control measures have been applied for international trade or disease prevention and control in a country or area” [18].

### 2.1. Semen Collection Centers

To better understand provisions of Reg. 686, it is important to focus attention on some aspects relating to the organization of semen collection centers and their biosecurity. Monaco and Lacalandra (2019) [4] provided a brief report on the abovementioned points, and highlighted several drawbacks regarding AI in dromedaries. These are mainly related to the semen collection process, such as the aggressive male reproductive behavior, and the type and quantity of ejaculate, characterized by high viscosity [2]. The male requires a long period of training, with the adoption of specific measures concerning both the equipment used and the preparation of technical personnel. It is also essential to maintain scrupulous conditions of cleanliness and wellbeing to prevent injuries resulting in infections of the genital organs, stress, and stereotypical behavior, which are also associated with poor reproductive performance.

Mansour et al., in 2022 [19], used a supersensitive glove inserted inside the female’s vaginal canal to collect sperm. After natural mating, the sperm deposited inside the glove by the male could be collected by pulling it out.

This technique proved effective in several respects. In fact, the males completely utilized the materials, and the mating course was not used.

The importance of the presence of a responsible veterinarian—as also indicated in the Terrestrial Animal Code [18]—and the division into different biosecurity zones (collection room, semen storage, laboratory, etc.) to allow physical separation between the various areas of the center, should be emphasized. 

In addition, the presence of a surveillance plan consistent with the principles of hazard analysis and critical control points (HACCP), along with the keeping of all documentation, must not be neglected.

The latter must provide clear evidence that biosafety, surveillance, and traceability practices are effectively applied [4].

The semen collection center should have a section dedicated to processing and storage operations [4]. Macroscopic examinations such as evaluation of volume, appearance, and consistency, along with microscopic examinations such as evaluation of concentration, mass motility, individual mobility, and morphological examination, should be carried out in the laboratory.

Camel sperm is known to have a high viscosity [3]. Since spermatozoa have a poor ability to move in their viscous matrix, dromedary spermatozoa must undergo total liquefaction before microscopic examination and before dilution [20]. However, the methods for performing liquefaction without damaging the spermatozoa have not been perfected [3,20]. Spontaneous liquefaction is not effective, as it results in partial liquefaction. Some of the most effective techniques to perform liquefaction include heating to 37 °C in a water bath placed on a magnetic stirrer, or the application of ultrasound [18].

The green egg yolk pad and Triladyl or the extenders OptiXcell and INRA 96 can be used in fresh semen as diluents [21]. If semen has to be used 24 h after collection, freezing is necessary. Unfortunately, to date, there are no approved and effective freezing/thawing strategies to maintain sperm viability, and this is a limitation to the application of artificial insemination on a large scale [20]. Freezing requires the addition of cryoprotectant agents, which may have potentially deleterious effects on spermatozoa. They must be used at an optimal dilution to obtain their benefits by avoiding osmotic and thermal shocks during freezing and thawing [20].

Camel semen is packaged in polypropylene or polyvinyl B straws [3]. As noted in Reg. 686, each straw must be marked with obligatory identifying information of the animal, the lot, and the center. If the semen is stored in pellets, the marks can be placed either on the pellets or on each other container where germ material is stored.

The pregnancy rate can be influenced by the number of inseminated sperm. The minimum effective insemination dose is 150 × 10^6^ sperm. It is possible to achieve higher pregnancy rates with fresh sperm by inseminating in the tip of the uterine horn, while chilled and frozen–thawed sperm show significantly lower fertility [21].

As far as artificial insemination is concerned, the most commonly used method of deposition, which results in a higher conception rate, is at the end of the uterine horn ipsilateral to the ovary, where the mature follicle is present, i.e., near the uterine–tubaric junction [20]. However, there is a paucity of literature concerning insemination protocols using refrigerated or frozen sperm [21]. Complications include estrus synchronization procedures and low or absent fertility of females inseminated with chilled or frozen sperm [3,21]. It is probably necessary to conduct studies on the screening of infectious diseases and the hygiene of sperm collection and processing [3,4,21].

#### Embryo Transfer

From the year 1990, the interest in embryo transfer has increased, especially with regard to improving genetics in camel dairy herds [20]. It has been demonstrated that the female dromedary has a long calving interval, producing about two calves in three years.

Nevertheless, the various reproductive inefficiencies known in this species can be circumvented by using multiple ovulation techniques and embryo transfer, while simultaneously optimizing mating plans with different males. The embryo transfer procedure requires superovulation of the donor female and synchronization of the recipients. Good pregnancy rates can be achieved with fresh embryos; however, an effective standardized method for embryo preservation needs to be formulated by refining freezing and thawing techniques. Thus, cryopreservation is a fundamental procedure that avoids the transport of live animals. There are two main methods for cryopreservation: slow cooling, and vitrification. In slow cooling, ethylene glycol is used as a cryoprotectant, and pregnancy rates of up to 37% have been achieved; however, this method is not practical under field conditions. Vitrification, on the other hand, proves to be much simpler and faster, and is therefore more widely applicable [20].

### 2.2. Animal Health Requirements

Article 38 of Reg. 686 lists the animal health requirements for movements to other MSs. Operators moving camelid germinal material must ensure that measures for the prevention and eradication of diseases transmissible to animals—which are covered in the AHL and updated in Regulation (EU) 1629/2018 [22]—are complied with, with the addition of the camelid diseases surra, infectious bovine rhinotracheitis (IBR), and epizootic hemorrhagic disease (previously known as deer hemorrhagic disease).

The criteria for the addition of diseases are highlighted in Article 5(3) of the AHL, and evaluation parameters for the establishment of the list of diseases are provided in Article 7.

According to Regulation (EU) 1882/2018 [23], diseases are subdivided into different categories. The subdivision criterion is related to “the different types of management measures, their potential severity and their impact on public or animal health, the economy, society or the environment”. “Management measures include basic responsibilities and obligations, such as reporting and notification of the detection or suspicion of a listed disease and eradication programs, as well as disease-specific, in-depth surveillance and eradication measures applicable throughout the Union, and measures relating to the movement of animals and products of animal origin into and within the Union”. “The systematic assessment by the Commission also took into account various factors, such as the species susceptible to certain listed diseases, disease reservoirs and disease vectors, whether or not the listed disease is currently present in the Union, and how the listed disease is transmitted between animals, and from animals to humans, and its potential impact on animal and human health, including its morbidity and mortality rates. The systematic assessment also considered the wider impact of these listed diseases, such as their impact on the economy, society, animal welfare, the environment and biodiversity”.

Category A diseases do not normally occur in the EU, and must be eradicated immediately; category B diseases are subject to control programs for eradication purposes, while category C diseases are only relevant for some MSs. Category D diseases require measures to prevent their entry into the EU and their subsequent spread between the MSs, while category E diseases only need to be monitored. All camelid diseases belong to categories D and E.

Relative to the animal species, epizootic diseases can be classified into different categories. For example, in camelids, brucellosis belongs to categories D and E, while in cattle and sheep and goats it is subject to control by eradication and, therefore, falls under category B.

The diseases of greatest concern in camels are reported below.

*Tuberculosis, Category D + E* [23].

Tuberculosis is a chronic zoonotic granulomatous disease caused by a group of related bacteria defined as the *Mycobacterium tuberculosis* complex (MTBC). Among these, the main pathogens belonging to MTBC isolated in camelids are *Mycobacterium tuberculosis*, *Mycobacterium bovis*, *Mycobacterium pinnipedii*, *Mycobacterium caprae*, and *Mycobacterium microti* [24].

In camels, the infection spreads mainly through the introduction of an infected animal into the herd, or because of close contact with infected animals of different species [25]. Transmission occurs primarily horizontally via aerosols from animals with pulmonary tuberculosis. Oral transmission can also occur through ingestion of contaminated food or water [25], and possibly also through wounds, urine, and feces [24]. Vertical congenital transmission is possible [24]. In farmed livestock, semen and vaginal secretions infected with tuberculosis act as transmission media, and pose a significant risk to susceptible animals and humans [26].

For the movement of camel germinal material into another MS, the animals from which it is taken must come from an establishment where the pre-movement surveillance program for infection with the *Mycobacterium tuberculosis* complex has been carried out for at least 12 months in accordance with Annex II Part 2 of Delegated Regulation (EU) 688/2020 [27]. Only animals that comply with the surveillance program may be brought into the establishment. The pre-movement surveillance program, therefore, applies to both the movement of animals and the movement of germinal material. For the movement of camelids, reference is also made to Article 23 of the abovementioned regulation. For germinal material, reference is made only to Annex II Part 2.

In particular, the pre-movement surveillance program for germinal material must therefore include the following:i.Postmortem inspection of all camels in the establishment where the animals are slaughtered.ii.Postmortem examination of all dead camels over 9 months of age, except where this is not possible for logistical or scientific reasons.iii.An annual animal health examination by a veterinarian.iv.Annual tests carried out, with negative results, on all camels kept in the establishment for breeding purposes.

By way of derogation, these annual tests are not necessary if:-The competent authority considers that the risk of infection is negligible in the MS or area where the establishment is located.-The surveillance program has been carried out for at least 24 months and no cases of infection have been reported during this period.-The establishment is situated in a state or zone free of infection in the relevant bovine population.

The annual tests that can be carried out according to Annex I of Delegated Regulation (EU) 2020/688 [27] are as follows:1.Intradermal tuberculin tests:
(a)IDT;(b)Comparative IDT.2.Tests available for blood samples:
(a)Interferon-gamma test.

If infection is suspected, investigations must be carried out to rule out the presence of the disease. If cases of infection with *Mycobacterium tuberculosis* have been reported in the establishment where the camels are kept, these animals may only be moved to another MS if all camelids over six weeks old in the establishment have been tested by blood samples, with negative results. These tests must be carried out at least 42 days after the removal from the establishment of the last confirmed case and the last positive animal.

*Brucellosis, Category D + E* [23].

Brucellosis is a zoonotic infectious disease of bacterial etiology caused by several species of the genus *Brucella*. Although camelids are not the primary hosts, they are susceptible to infection by the three *Brucella* species with the greatest economic and zoonotic impact [28,29]—namely, *B. abortus*, the primary agent of bovine brucellosis; *B. melitensis*, the agent of caprine brucellosis; and *B. suis*, the etiological agent causing brucellosis in pigs [28].

Camels become infected mainly through direct mucosal contact with lochia, aborted fetuses, and placentae, but also through contaminated fragments of sand transported from a distance [30]. Infection is also possible through contaminated pastures, feed, or water. The disease is mostly found in intensively reared camels, and cohabitation with other farm animals is considered to be a risk factor [25,31]. The disease manifests in a chronic form, and may not even cause obvious clinical signs. Abortion—in the second half of pregnancy—and stillbirth frequently occur. Other disorders such as placental retention, granulomatous endometritis, hydrobursitis, ovariobursal adhesions, and mastitis may also occur. Visceral abscesses, hygroma, arthritis, and lameness have also been reported [25]. In male camels, *B. abortus* and *B. melitensis* have been associated with orchitis and epididymitis [29].

To prevent the spread of brucellosis, appropriate management, health, and biosecurity measures must be implemented. Animals must be regularly tested for brucellosis, and those that test positive must be eliminated.

Any female that aborts must be promptly isolated and analyzed to determine the cause of the abortion. Samples from aborted fetuses and fetal membranes must be submitted for laboratory diagnosis, and the rest must be disposed of properly. It is also important that all agricultural workers observe personal hygiene, such as wearing protective clothing and washing their hands regularly, to protect themselves and prevent contamination.

For the diagnosis of brucellosis, the most effective method is the culture and isolation of the pathogen, followed by bacteriological testing and biotyping [18].

For the movement of camelid germinal material to another MS, the animals from which it is taken must come from an establishment where no cases of infection with *Brucella abortus*, *melitensis*, or *suis* have been reported for at least 42 days prior to the date of collection.

All dromedaries in the establishment must also undergo a serological test for infection with *Brucella abortus*, *melitensis*, and *suis* 30 days prior to the collection of germinal material.

The serological tests are listed in Annex I Part 1 of Delegated Regulation (EU) 688/2020 [27]:-Buffered *Brucella* antigen test;-Complement fixation test;-Indirect ELISA;-Polarized fluorescence method;-Competitive ELISA.

*Infectious Rhinotracheitis, Category D + E* [23].

Infectious rhinotracheitis/infectious pustular vulvovaginitis is a disease caused by bovine herpesvirus type 1. It belongs to the family Herpesviridae, subfamily Alphaherpesvirinae. It is one of the main contagious infectious diseases of cattle, as it causes severe economic damage despite its low mortality rate, mainly due to latent infections. This has also been found in camels, where a direct pathogenic effect in respiratory infections has been confirmed [32]. The seroprevalence of BoHV-1 infection has also been reported in camels, and the introduction of newly acquired animals into the herd is considered to be the most important risk factor related to seropositivity [33].

Despite little knowledge of the pathogenic effects of this virus in dromedaries, this species is dangerous because it can act as a reservoir and, therefore, as a passive player in the spread of the virus to other susceptible animal species [33].

Reg. 686 states that for the movement of camelid germinal material to another MS, the animals from which it is taken must come from an establishment where no case/confirmed infection of infectious bovine rhinotracheitis/infectious pustular vulvovaginitis has been reported for at least 30 days before the date of collection of the germinal material.

*Rabies, Category B + D + E* [23].

Rabies is an encephalomyelitic zoonotic disease classified by the OIE as lethal. The incubation period of the disease is variable, and is nearly 100% fatal after the onset of clinical symptoms in both animals and humans in the absence of adequate post-exposure prophylaxis. Like all warm-blooded animals, camels are susceptible to rabies, and the prophylactic measure to be implemented—as recommended by the OIE—is preventive vaccination in both companion and farm animals, as well as in wild animals. This involves the use of an inactivated vaccine [25].

Reg. 686 states that for the movement of camelid germinal material to another member state, the animals from which it is taken must come from an establishment where no case/confirmed infection of rabies virus has been reported for at least 30 days before the date of collection of the germinal material.

*Anthrax, Category D + E* [23].

Anthrax is a serious contagious disease with worldwide distribution, which mainly affects herbivores, but can also affect other animals, including humans. The etiological agent is *Bacillus anthracis*—a sporogenous, aerobic, Gram-positive bacterium. Spore formation occurs in the presence of oxygen, and spores can survive in the soil for several decades.

Methods of infection of herbivorous animals can be traced back to ingesting spores from contaminated soil and plants while grazing or drinking contaminated stagnant water. Transmission can also occur via flies that have fed on infected individuals or carcasses—mostly from the genera *Hippobosca* and *Tabanus*, but also including *Cephalopina titillator* larvae. Other sources of infection could include migratory birds. The disease has a rapid progression, with an acute or hyperacute septicemic course, so any attempt at antimicrobial therapy with penicillin or tetracycline proves to be of little value [25]. Carcasses of animals that have died of anthrax, along with their bedding and dung, must be incinerated or buried deep in the ground with quicklime. A postmortem examination is not indicated, so as to avoid the dispersal of spores in the environment. Contaminated utensils must be disinfected using strong disinfectants such as sodium hydroxide (10%), formaldehyde (5%), hydrogen peroxide (7%), or glutaraldehyde (2%).

For vaccine prophylaxis in dromedaries, the Sterne vaccine is available, which is a live, avirulent spore vaccine, and confers immunity for about 9 months.

Affected premises must remain in quarantine until vaccination is completed. It is important to apply strict sanitary measures, especially when it comes to rodent and insect control. Anthrax is a notifiable disease [25].

Reg. 686 states that for the movement of camelid germinal material to another MS, the animals from which it is taken must come from an establishment where no case/confirmed infection of anthrax has been reported in at least the last 15 days before the date of collection of the germinal material.

*Epizootic hemorrhagic disease, Category D + E* [23].

Epizootic hemorrhagic disease (EHD) is a non-contagious, arthropod-borne viral disease that affects some wild ungulates, such as white-tailed deer and, rarely, cattle [34,35]. The EHD virus (EHDV) is closely related to bluetongue virus, both belonging to the genus *Orbivirus* in the family Reoviridae, and having the same mode of transmission—namely, through the bites of midges belonging to the genus *Culicoides*. EHDV strains have been classified into seven serotypes [25]. It is assumed that the epidemiological characteristics mirror those of bluetongue virus, encompassing tropical and temperate regions. Positive serological evidence has been documented in camels. EHD is a lethal disease in deer. Lesions in deer reflect diffuse vascular damage and subsequent disseminated intravascular coagulation. In cattle, episodes with high mortality have been reported, but the virus often occurs in a subclinical form [36].

Reg. 686 states that measures must be observed with respect to epizootic hemorrhagic disease for the movement of germinal material of camels. The animals must come from an establishment that has reported no infections with epizootic hemorrhagic disease virus within a radius of at least 150 km for a period of at least two years from the date of collection of the germinal material.

*Surra, Category D + E* [23].

Surra is a parasitic infectious disease caused by *Trypanosoma evansi*. The pathogen is mechanically transmitted by hematophagous flies of the genera *Stomoxys* and *Tabanus*, and has no developmental stage in its vector. Transmission is facilitated if the blood meal transfer from one animal to another occurs within a short period of time, as the infectious capacity depends on the survival time of the trypanosome within the oral cavity of the vector [37]. The species in which surra has been reported include camels—both dromedary and Bactrian—equines, cattle, sheep, buffalo, dogs, cats, and pigs [18,38].

In dromedaries, surra can occur either in an acute form (which can last up to several months)—consisting of high fever, anemia, weakness, and death [39]—or in a chronic form. The latter is encountered more frequently, and can last up to several years [39]. It is characterized by intermittent fever, low milk and meat production, an increase in abortions, diffuse subcutaneous edema in the ventral and scrotal areas, progressive anemia, blindness, lethargy, neurological disorders, and altered hemostasis manifested by petechiae and ecchymosis. High mortality is also found in breeding [37,39]. Abortion probably occurs because of placentitis and placental edema. Premature births have also been observed in dromedaries [29]. In male dromedaries, infection with *Trypanosoma evansi* is associated with severe testicular degeneration caused by fever attacks and the formation of immune complexes that are deposited at the level of the testes, impairing the function of Sertoli cells. There is also pathogenic action on the pituitary gland in both sexes, which in the male can contribute to testicular degeneration and, in the pregnant female, compromises fetal development. Testicular pathology is reversible after treatment but, in severe cases, it can become permanent [29].

In some cases—and especially in species other than camelids—the disease can manifest itself in a non-serious form, and can lead to immunosuppression, which then results in increased susceptibility to other infections, failure of vaccination campaigns for other infectious diseases and, consequently, severe economic losses [37]. The severity of the syndromes found depends on the virulence of the strain and the susceptibility of the host. Acute forms with high mortality are mainly found in virgin populations. In enzootic areas, on the other hand, animals may possess resistance, and more often manifest a chronic or subclinical form [39]. Clinical signs are indicative but not pathognomonic, and must therefore be confirmed by laboratory methods [18]. Prevention can be implemented through the administration of the trypanocide melarsomine—which is safe during pregnancy—during the season most at risk [29].

This disease was not considered for many years within the animal health conditions in trade in the EU, but after a series of outbreaks occurred in the Canary Islands and then spread to Spain and France, surra was included in Regulation (EU) 429/2016 [37].

Delegated Regulation (EU) 688/2020 [27] states that for movements of dromedary germinal material to another MS, the animals from which it is taken must come from an establishment where no case of surra has been reported for at least 30 days prior to the date of collection of the germinal material. If the disease has been confirmed in the previous two years since the last outbreak, the establishment must have remained under restrictions. These restrictions have a minimum duration of 6 months, since serological tests (with negative results) to exclude the presence of the disease must be carried out at least 6 months after the infected animals have been removed.

The tests to be carried out, according to Annex I Part 3 of Delegated Regulation (EU) 688/2020 [27], are as follows:-ELISA for trypanosomiasis;-Paper agglutination test for trypanosomiasis, with a serum dilution of 1:4.

*Bluetongue (Serotypes 1–24), Category C + D + E* [23].

Bluetongue (BT) is a subacute or acute viral disease caused by the bluetongue virus (BTV) of the genus *Orbivirus* and family Reoviridae, mainly transmitted by hematophagous insect vectors of the genus *Culicoides*, within which viral replication is possible. Iatrogenic transmission with contaminated needles, vertical transmission (only for serotype 8), and transmission via the semen of viremic bulls have also been demonstrated [18]. To be infectious, the virus must migrate from the intestines to the salivary glands of the vector insect. This process, which makes the midges effective vectors of the virus, takes place about 10 days after the blood meal [36].

There are 27 recognized serotypes of BTV. The main vertebrate hosts can include wild and domestic ruminants, camelids, and wild canids. However, the species most susceptible to developing clinical symptoms are sheep, and among the breeds of this species, European breeds are more susceptible than subtropical or tropical breeds [18].

In Europe, the occurrence of BT is directly related to the peak of vector activity, coinciding with the seasonal increase in temperatures from summer to early autumn [18]. BT virus has been isolated from the blood of experimentally infected dromedaries. Although they do not develop clinical signs, for serotypes 1 and 8, the presence of the virus in the blood, seroconversion, and the development of neutralizing antibodies have been demonstrated. Therefore, this animal could play a role in transmission to the local *Culicoides* midge population—a source of infection for small and large ruminant herds living in neighboring areas [40]. Wild or extensively reared ruminants may be particularly important for the persistence of the virus’ circulation in a region [18].

Therapeutic treatment of the disease is non-specific, directed towards alleviating symptoms, and is logistically demanding and unrewarding [36].

Regarding prophylactic measures, the infection can be prevented through the removal of midges from the environment, although in endemic areas or extensive environments this is not feasible [36]. Vaccination, on the other hand, is more suitable in areas where BT is endemic, but also in areas where the disease is first found epidemically. Inactivated vaccines are safer than live, attenuated vaccines [36]. To achieve effective protection, animals should be vaccinated for all BTV serotypes present in the region. The vaccines should be administered before the period when the virus is present in the region, and before the beginning of the breeding season, so as to avoid possible teratogenic effects on the fetus [36].

Reg. 686 refers to the requirement for donor animals to comply with the animal health requirements in Annex II Part 5 Chapter 2. Semen donors and donors of oocytes and embryos are considered separately in the regulation.

For the movement of sperm to another MS, at least one of the following conditions must be met:1.Donors have been held in a bluetongue-free state or zone for at least 60 days prior to collection.2.Where dromedaries are held in a bluetongue-free zone during the 60 days prior to the collection during the bluetongue-free season, an approved bluetongue eradication program exists. If no such eradication program exists in the MS or in the area where the establishment is located, the competent authority of the place of destination itself may give its consent to the conditions of an establishment in a seasonally free zone and accept the consignment of semen.3.Donor animals are kept in a vector-protected establishment for at least 60 days after collection.4.Donor animals are subjected to a serological test for antibodies to serogroups 1–24, with negative results, between 28 and 60 days before the date of each semen collection.5.Donor animals are subjected to a bluetongue virus agent identification test (serotypes 1–24) carried out on blood samples taken at the beginning and end of the semen collection, with negative results.
-For virus isolation, test at intervals of at least 7 days.-For PCR, test at intervals of at least 28 days.

Semen used to fertilize oocytes must also follow these requirements.

Regarding donors of embryos conceived in vivo and oocytes for in vitro embryo production, the requirements are similar to those for sperm donors, but the serological tests described Point 4 and the viral agent identification tests described in Point 5 must be carried out on a blood sample taken on the day of oocyte or embryo collection.

#### 2.2.1. Health Certification for Exchanges of Germinal Products between Member States

Regarding animal health certification, this is defined in Article 161 of the AHL for cattle, sheep, goats, pigs, and horses. When this is required for other animals, it must still follow the requirements of Articles 161 and 162. Its use is therefore mandatory in the case of trade between MSs, and possibly within the same MS, if disease control measures under Article 55 (if one of the categories of disease is suspected) and emergency measures under Articles 257 and 258 are in place.

Article 39 of Reg. 686 stipulates that the health certificate for germinal material of animals of the *Camelidae* family must be signed by an official veterinarian after conducting a visual examination of the transport container and a documentary check of the data submitted by the operator.

The information contained in the health certificate can be found in Annex IV, point 2 of Reg. 686. Templates for health certificates are available in Regulation (EU) 2235/2020 [41].

Interestingly, the health certificate for germinal material of camels does not indicate the presence of antibiotics in diluents. In cattle, small ruminants, and pigs, the OIE suggests hygienic ways of collecting and handling semen to prevent the likelihood of contamination by potentially pathogenic common bacteria.

Therefore, in Reg. 686, it is noted that the presence of antibiotics in seminal diluents must also be indicated in the health certificate. This indication, however, is not present in the health certificates for germinal material of camels, and could compromise the hygiene of the material itself and the prudent use of antibiotics. It should also be noted that these animals can be susceptible to important infections [42] that are explicitly mentioned in the regulation, for which antibiotics should be specifically used in seminal diluents, and which are associated with infertility in the dromedary camels [42], such as *Mycoplasma* and *Leptospira* [18]. *Mycoplasma* has been identified in dromedary camels’ vagina, cervix, uterus, foreskin, and semen samples [25,42]. Leptospira, on the other hand, has been detected in semen samples from infertile camels, but its presence has not been directly associated with infertility, as it may be attributable to contamination of the urine of chronically infected animals that harbor the pathogen in their kidneys [42]. To date, however, it is uncertain whether camels are vulnerable to leptospirosis, as clinical symptoms have not yet been described in this species, but have been isolated in aborted fetuses [25,42].

#### 2.2.2. Information Management System for Official Controls Related to Germplasm Movement within the EU

In order to notify when consignments of germinal products are intended to be moved to other MSs, the TRACES system should be used. However the functionalities of this system are integrated with an information management system for official controls (IMSOC). This—as stated in Article 2, point 28, of Reg. 686—is dedicated to the integrated functioning of the mechanisms and tools “*through which data, information and documents concerning official controls and other official activities are managed, handled, and automatically exchanged*”.

There are procedures of emergency for the notification of movements of consignments of germinal products within the EU; in situations of power cuts and/or other disturbances of IMSOC (Article 43), the competent authority of the place of origin of the consignment of these products “[…] *shall notify the Commission and the competent authority of the place of destination of the movement of that consignment by fax or email”*. The same procedure is applied for germinal products intended for scientific purposes or for delivery to gene banks in events such as those described above (Article 48).

## 3. Discussion

European legislation on animal breeding—including that of camels—has also contributed to the conservation of animal genetic resources, the protection of genetic biodiversity, and the production of typical regional high-quality products based on the specific hereditary characteristics of local domestic animal breeds. Indeed, it has promoted sound breeding programs for the improvement of breeds, their preservation, and the conservation of genetic diversity.

These established procedural rules also seek to contain the circulation of infectious animal diseases—both contagious and non-contagious—that can spread between the MSs. Germinal material—especially semen—is most incriminated as a vector for the spread of infectious animal diseases, due to its inherent nature.

The aim of Reg. 686 is to limit the risk of transmission of diseases through AI and other assisted reproductive technologies. Semen must be collected and processed in approved and controlled semen collection centers, obtained from animals whose health status guarantees that there is no risk of spreading diseases, and collected, processed, stored, and transported according to rules that preserve their health status.

Since camels are present in Europe in negligible numbers—certainly not comparable to those of other farm animals—they are not subject to the same conditions, because they are less at risk of spreading communicable diseases.

However, since they are susceptible to various infections with contagious diseases belonging to category A (i.e., those that do not normally occur in the EU, and which require immediate eradication measures as soon as they are detected), common to several species of domestic ungulates—such as foot-and-mouth disease or rinderpest—it is considered that they should be better monitored in this respect.

It is considered important to emphasize that establishments where germinal material is processed together with that of other breeding species could also be regulated, in order to make a possible development of the camelid breeding sector and the consequent exchange of germinal material in the EU more effective, with a view to the trend towards more sustainable and diversified animal production—especially in those areas of the EU where semi-arid pasture areas exist or may develop in the future.

An analysis of the impacts on animal welfare should also be carried out alongside biotechnological research.

## 4. Conclusions

Reg. 686 is the first European law to harmonize the rules on the movement of camel germplasm within the EU. Although this regulation is complex, it provides important technical indications and the minimal contents of health certification. More traceability is also needed for germplasm movements, where veterinary services exercise strict controls. The importance of this legislation and its associated regulations is related to the control of the germinal products’ movements and, of course, to the reduction in infectious diseases to improve animal welfare and increase production. Therefore, in conclusion, the cornerstone of this regulatory framework is based on “safe”.

## Figures and Tables

**Table 1 animals-12-02255-t001:** Articles of Regulation (EU) 2020/686 relating to movements of germinal products of camelids to other member states.

Article	Topic
Article 11	Traceability requirements
Article 38	Animal health requirements
Articles 39 and 40	Rules concerning animal health certification
Articles 41, 42, and 43	Rules on notification
Articles 3 and 48	Emergency procedures for the notification of movements of consignments of germinal products

## Data Availability

Not applicable.

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
