# Peer review of "European Regulations on Camel Germplasm Movement within the European Union: A Current Framework Based on Safety"

_animals, 2022, doi:10.3390/ani12172255_

Round 1

Reviewer 1 Report

Before the promulgation of such regulation, there was no mention of the possibility of Camelidae germplasm movement and its zootechnical use” within EU countries, therefore the analysis of such new regulation and of its most important parts might be of great benefit for veterinarians and breeders that wish to sell their high-quality germinal material or to genetically improve their herds

However there are some aspects that the authors did not take into account during the preparation of the work: the study has focused on dromedary camel germinal material, mainly semen. Despite big progress has been made regarding artificial insemination in this species, such assisted reproductive technology is still far from giving the results that might be obtained with the transfer of chilled or frozen-thawed embryos, also considering the impact of such technology in speeding up the genetic improgement.

It is therefore strongly suggested that, even considering the possibilities of a future use of frozen semen, the work will take into account aspects related to the embryo transfer technology.

In addition, other species of Camelidae (i.e. Llama, Alpaca, and two-humped camels) are currently  bred in Europe and probably the number of heads is higher than dromedary camels; therefore it would be useful if authors could report the number of such species in Europe (so as to underline the impact of the germinal material use) and briefly mention about assisted reproductive technologies also in such species.

By considering those two aspects, the impact of the manuscript will be certainly improved and it will be of great benefit in promoting and assisting the use of assisted reproductive technologies in Camelidae species within the European territory.

Finally, it might be of use, according to the authors’ choice, if the manuscript could have a more practical approach: what camel breeders and their veterinarians shall do, in terms of regulation, for the collection, transportation, and use of their camel germinal material (either semen or embryos)?

Below there are some other remarks:

 Line 13-15: the cited regulation does not mention about the commerce of breeding animals, nor about the importation of germinal material into the EU.

Lines 15-16: “Due the camel breeding industry is set to grow strongly, an important amount of germinal material of terrestrial animals, including they are belonging of Camelidae’ family”: there are no reports about the movement of such “important amount” of genetic material within the EU. Moreover, the sentence seems not grammatically correct even if it is taken from the regulation itself.  

Line 20: please replace the word “legislature” or rephrase the sentence

Lines 26-28: What does the sentence “EU with its regulations is preventing the arrival through these animals of emerging or even no longer present diseases in Europe” means? Do the authors refer to the fact that the entrance of genetic material from extra EU countries is prevented? Please explain

Lines 35-36: Please add the word “breeders” to the following sentence “Camel breeding has always played an important role in world agriculture and is now also becoming a source of income for the European Union (EU) BREEDERS.

Lines 36-38: the sentence “ It is necessary to promote ….” is not clear: what do the authors refer to regarding “performance standards”?

In addition, the word “standard is repeated three times in three lines, please correct this.

Lines 40-43: please refer to the specific law or regulation related to animal importation into EU or correct the sentence.

In lines 47-53, reproductive seasonality is not mentioned as a contributing factor to the low reproductive efficiency. Please replace “high mortality of the young” with “high neonatal mortality”.

The fact that diagnostic procedures and investigations related to infertility are not routinely implemented according to standard procedures might also be a contributing factor

Lines 54-56: Unfortunately AI is not yet routinely used for breeding in the middle east, contrarywise Embryo transfer is.  

116-118: what does this sentence mean?

Lines 119-146  This paragraph is not clear: Please define clearly the difference between different types of establishments of germinal material. Where and how camel semen or embryo could be collected?

Lines 147-202: Please delete this paragraph. Most of the information provided in this paragraph has been already reported in a detailed manner in the cited article (i.e. Monaco and Lacalandra 2020) moreover the information reported about semen processing is not updated and the one related to semen freezing and artificial insemination is in contrast with the statement reported in the introduction (use of AI in Middle East etc).

333-340 have the authors found references about the clinical manifestation of BoHV-1 in camels? Seroprevalence has only been demonstrated. Please verify

407 please refer to the work of Gutierrez et al., 2010 related to Trypanosomiasis in Dromedary camels.

The manuscript does not mention the IMSOC system for the official control of information related to germplasm movement within the EU 

Author Response

Dear Reviewer,

Thank you very much for your time and all your comments.

We have revised the manuscript considering your comments; the answers to your questions are given below.

The changes made in the manuscript to address comments are written in red.

The manuscript has been slightly improved; however, some contradiction between results and discussion are still present and the conclusion is not completely supported by your results. Please, read the comments carefully and modify accordingly or explain your opinion in the reply to the comment.

Before the promulgation of such regulation, there was no mention of the possibility of Camelidae germplasm movement and its zootechnical use” within EU countries, therefore the analysis of such new regulation and of its most important parts might be of great benefit for veterinarians and breeders that wish to sell their high-quality germinal material or to genetically improve their herds

However there are some aspects that the authors did not take into account during the preparation of the work: the study has focused on dromedary camel germinal material, mainly semen. Despite big progress has been made regarding artificial insemination in this species, such assisted reproductive technology is still far from giving the results that might be obtained with the transfer of chilled or frozen-thawed embryos, also considering the impact of such technology in speeding up the genetic improgement.

It is therefore strongly suggested that, even considering the possibilities of a future use of frozen semen, the work will take into account aspects related to the embryo transfer technology.

  1. The embryo transfer technology has been included in the text.

In addition, other species of Camelidae (i.e. Llama, Alpaca, and two-humped camels) are currently  bred in Europe and probably the number of heads is higher than dromedary camels; therefore it would be useful if authors could report the number of such species in Europe (so as to underline the impact of the germinal material use) and briefly mention about assisted reproductive technologies also in such species.

  1. We do not consider it appropriate to report the part of the regulation relating to llama and alpaca because the MS would become exceedingly long. We would take into consideration these species in another review. 

Relating to the number of camelids in Europe there are no official data, but only numbers available on informative and non-institutional websites. For example,  https://camel-milk.org/publications/ (… are increasingly present around the Mediterranean Basin including in the European agricultural landscape…).

The impact and the increasing number of germinal materials used in camelids are underlined in the Recital no. 23 of Regulation EU 2020/686: “An increasing number of germinal products of dogs and cats, of terrestrial animals other than bovine, porcine, ovine, caprine and equine animals kept at confined establishments and of animals of the families Camelidae and Cervidae are moved between the Member States. Therefore, it is appropriate to establish harmonized rules on the marking of straws and other packages containing such germinal products. Additional rules on the traceability of germinal products of kept terrestrial animals of species other than those of the bovine, porcine, ovine, caprine, and equine species should be laid down in this Regulation”.

By considering those two aspects, the impact of the manuscript will be certainly improved and it will be of great benefit in promoting and assisting the use of assisted reproductive technologies in Camelidae species within the European territory.

Finally, it might be of use, according to the authors’ choice, if the manuscript could have a more practical approach: what camel breeders and their veterinarians shall do, in terms of regulation, for the collection, transportation, and use of their camel germinal material (either semen or embryos)?

Below there are some other remarks:

 Line 13-15: the cited regulation does not mention about the commerce of breeding animals, nor about the importation of germinal material into the EU.

  1. The sentence has been modified.

Lines 15-16: “Due the camel breeding industry is set to grow strongly, an important amount of germinal material of terrestrial animals, including they are belonging of Camelidae’ family”: there are no reports about the movement of such “important amount” of genetic material within the EU. Moreover, the sentence seems not grammatically correct even if it is taken from the regulation itself.  

  1. The sentence has been modified.

Line 20: please replace the word “legislature” or rephrase the sentence,

  1. The sentence has been rephrased.

Lines 26-28: What does the sentence “EU with its regulations is preventing the arrival through these animals of emerging or even no longer present diseases in Europe” means? Do the authors refer to the fact that the entrance of genetic material from extra EU countries is prevented? Please explain

  1. The sentence has been modified and clarified.

Lines 35-36: Please add the word “breeders” to the following sentence “Camel breeding has always played an important role in world agriculture and is now also becoming a source of income for the European Union (EU) BREEDERS.

  1. The sentence has been modified.

Lines 36-38: the sentence “ It is necessary to promote ….” is not clear: what do the authors refer to regarding “performance standards”? In addition, the word “standard is repeated three times in three lines, please correct this

  1. The sentence has been modified, and the word standard has been changed.

Lines 40-43: please refer to the specific law or regulation related to animal importation into EU or correct the sentence.

  1. Delegated Regulation (EU) 2020/686 (hereafter Reg. 686) has been specified as reference required.

In lines 47-53, reproductive seasonality is not mentioned as a contributing factor to the low reproductive efficiency. Please replace “high mortality of the young” with “high neonatal mortality”.

  1. “High mortality of the young” has been modified with “high neonatal mortality”.

The fact that diagnostic procedures and investigations related to infertility are not routinely implemented according to standard procedures might also be a contributing factor

  1. The sentence has been modified.

Lines 54-56: Unfortunately AI is not yet routinely used for breeding in the middle east, contrarywise Embryo transfer is.  

  1. The sentence has been modified.

116-118: what does this sentence mean?

  1. The sentence has been modified and clarified.

Lines 119-146  This paragraph is not clear: Please define clearly the difference between different types of establishments of germinal material. Where and how camel semen or embryo could be collected?

  1. The paragraph has been modified and the difference between different types of establishments of germinal material has been clarified.

Lines 147-202: Please delete this paragraph. Most of the information provided in this paragraph has been already reported in a detailed manner in the cited article (i.e. Monaco and Lacalandra 2020) moreover the information reported about semen processing is not updated and the one related to semen freezing and artificial insemination is in contrast with the statement reported in the introduction (use of AI in Middle East etc).

  1. The paragraph has been deleted.

333-340 have the authors found references about the clinical manifestation of BoHV-1 in camels? Seroprevalence has only been demonstrated. Please verify.

  1. The information has been verified and the sentence has been corrected.

407 please refer to the work of Gutierrez et al., 2010 related to Trypanosomiasis in Dromedary camels.

  1. The reference has been included.

The manuscript does not mention the IMSOC system for the official control of information related to germplasm movement within the EU 

  1. This aspect has been included in a paragraph.

Reviewer 2 Report

The authors performed an analysis of the Delegated Regulation (EU) 2020/686 (hereafter Reg. 686) about the traceability and animal health for the movement of germinal material of camels within the EU.

I am not familiar with this type of review and I do not know if it may be of interest to the readers of Animals.

I would suggest including how the literature was searched and screened. I would increase the number of references related to camel spermiology and their handling in the training center by referring to more references (better if primary reference) and not only to the review of Monaco and Lacalandra.

Considering that embryo transfer is quite common and the work on embryos is increasing, I would suggest considering also the possibility of shipping embryos and not only semen.

Moreover, it would be good to consider the dromedary camel population in Europe and describe the quantity and type of farms, possible movements of animals, and their products within and across Europe. Similarly, for the disease, the incidences of the cited disease in the different dromedary camel populations (e.g. North Africa, Middle East etc) should be included to understand the epidemiological scenario we are discussing.

English needs attention particularly in the summary and in the abstract.

The authors performed an analysis of the Delegated Regulation (EU) 2020/686 (hereafter Reg. 686) about the traceability and animal health for the movement of germinal material of camels within the EU.

I am not familiar with this type of review and I do not know if it may be of interest to the readers of Animals.

I would suggest including how the literature was searched and screened. I would increase the number of references related to camel spermiology and their handling in the training center by referring to more references (better if primary reference) and not only to the review of Monaco and Lacalandra.

Considering that embryo transfer is quite common and the work on embryos is increasing, I would suggest considering also the possibility of shipping embryos and not only semen.

Moreover, it would be good to consider the dromedary camel population in Europe and describe the quantity and type of farms, possible movements of animals, and their products within and across Europe. Similarly, for the disease, the incidences of the cited disease in the different dromedary camel populations (e.g. North Africa, Middle East etc) should be included to understand the epidemiological scenario we are discussing.

English needs attention particularly in the summary and in the abstract.

Author Response

Dear Reviewer,

Thank you very much for your time and all your comments.

We have revised the manuscript considering your comments; the answers to your questions are given below.

The changes made in the manuscript to address comments are written in red.

The manuscript has been slightly improved; however, some contradiction between results and discussion are still present and the conclusion is not completely supported by your results. Please, read the comments carefully and modify accordingly or explain your opinion in the reply to the comment.

The authors performed an analysis of the Delegated Regulation (EU) 2020/686 (hereafter Reg. 686) about the traceability and animal health for the movement of germinal material of camels within the EU.

I am not familiar with this type of review and I do not know if it may be of interest to the readers of Animals.

I would suggest including how the literature was searched and screened. I would increase the number of references related to camel spermiology and their handling in the training center by referring to more references (better if primary reference) and not only to the review of Monaco and Lacalandra.

Considering that embryo transfer is quite common and the work on embryos is increasing, I would suggest considering also the possibility of shipping embryos and not only semen.

  1. The embryo transfer technology has been included in the text.

Moreover, it would be good to consider the dromedary camel population in Europe and describe the quantity and type of farms, possible movements of animals, and their products within and across Europe. Similarly, for the disease, the incidences of the cited disease in the different dromedary camel populations (e.g. North Africa, Middle East etc) should be included to understand the epidemiological scenario we are discussing.

  1. Relating to quantity of camelids in Europe there are not official data, but only numbers available on informative and non-institutional websites. For example, https://camel-milk.org/publications/ . The sentence in which is reported that there is an increasing number of the germinal material use in camelids is cited in Recital no. 23 of the Regulation EU 2020/686

English needs attention particularly in the summary and in the abstract.

  1. English language has been reviewed.

Round 2

Reviewer 1 Report

The manuscript has been improved but some comments have not been properly addressed:

Regarding the number of camel heads in Europe, authors might find some important information by reviewing some studies/reports of Faye B.

According to the manuscript and 686 regulation a Camelidae (?) semen/embryo collection centre seems to be the only chance for the collection of camel germplasm, therefore a short description of south American camelid-assisted reproductive technologies could be provided (i.e. Semen and Embryo collection). One paragraph would not affect the manuscript length since another paragraph was significantly shortened.

Hereby there are further remarks:

Lines 26-27: The sentence is confusing. Please be consistent with the objective of the study:  the manuscript and the considered regulation take into consideration the germplasm movement within EU countries

Line 39-42: “ European legislator issuing delegated Regulation (EU) 2020/686 (hereafter Reg. 686) has established rules on zootechnical and genealogical conditions for trade in breeding animals, including camelid species, their germinal products (semen, oocytes, and embryos), and their imports into the EU.”

As before, this sentence is confusing: regulation 686 does not take into account genealogical conditions for trade in breeding animals nor importation of camels into the EU. If other regulations take into account those aspects they must be clearly indicated; the sentence must be modified.

Lines 54-55This sentence is not correct “it has been only recently employed, as routine, to improve genetic traits such as milk production, meat, wool and the ability to compete in the Middle East [2] [3]”. The technique is not employed as routine, please change the sentence indicating that the technique is under development for its possible/potential use, on a large scale, for the indicated objectives.

Line 142 “2.1.2. Embryo transfer”; where is paragraph 2.1.1.?

Apologize for not being enough detailed. The following information might be useful so as to provide a quick idea about the context. Authors might report a short summary about the issue of semen collection centres and its critical aspects:

“To better understand provisions of the Reg. 686, it is important to focus attention on some aspects relating to the organization of semen collection centre and points about the biosecurity. Monaco and Lacalandra (2019) [4] provided a brief report on the above-mentioned points, and they have highlighted several drawbacks regarding AI in the dromedary.

The importance of the presence of the responsible veterinarian, as also indicated in the Terrestrial Animal Code [13], and the division into different biosecurity zones (collection room, semen storage, laboratory, etc.) to allow physical separation between the various areas of the centre is emphasised. In addition, the presence of a surveillance plan consistent with the principles of hazard analysis and critical control points (HACCP) and the keeping of all documentation is not negligible. The latter must provide clear evidence that biosafety, surveillance, and traceability practices are effectively applied [4].

Unfortunately, to date, there are no approved and effective freezing/thawing strategies to maintain sperm viability, and this is a limitation to the ap-182 plication of artificial insemination on a large scale [14].

It is probably necessary to conduct studies on the screening of infectious diseases and the hygiene of sperm collection and processing [4] [3] [15].

Moreover: the author did not address a previous comment about where camel semen could be collected (confined establishment, recognized establishment?) Please clarify the difference in a practical approach base.

i.e. for Sheep and Goats semen can also be collected in a confined establishment if certain conditions are assumed (please refer to the related regulation articles): Authors shall report such circumstances and point out that, despite not being considered by the law it is quite difficult that a camel semen collection centre could be built in Europe (cost, animal movements, little benefit due to the low number of Camelidae heads) therefore exception such as the one for rams and bucks shall probably be considered. 

Author Response

Dear Reviewer,

Thank you very much for your time and all your comments.

 We thank for your precise and thoughtful comments and constructive criticism, which has led to a better manuscript.

We revised the manuscript in relation to the suggestions and more detailed answers are given below.

The changes made in the manuscript to address comments are marked up using the 
“Track Changes” function.

Q: Regarding the number of camel heads in Europe, authors might find some important information by reviewing some studies/reports of Faye B.

A: In previous letter we have specified that the official number of camelids cannot be determined exactly. The same consideration is reported by Faye B. (2020). He states that often the number is not updated and is few reliable. However, at the end of the introduction we have added a sentence relating to your request (lines 74-77).

Q: According to the manuscript and 686 regulation a Camelidae (?) semen/embryo collection centre seems to be the only chance for the collection of camel germplasm, therefore a short description of south American camelid-assisted reproductive technologies could be provided (i.e. Semen and Embryo collection). One paragraph would not affect the manuscript length since another paragraph was significantly shortened.

A: We have added a paragraph 2.1.1 relating to the semen collection center. About embryo transfer the paragraph had already been added (lines 176-236).

Q: Line 39-42: “ European legislator issuing delegated Regulation (EU) 2020/686 (hereafter Reg. 686) has established rules on zootechnical and genealogical conditions for trade in breeding animals, including camelid species, their germinal products (semen, oocytes, and embryos), and their imports into the EU.” As before, this sentence is confusing: regulation 686 does not take into account genealogical conditions for trade in breeding animals nor importation of camels into the EU. If other regulations take into account those aspects they must be clearly indicated; the sentence must be modified.

A: We have modified the sentence (lines 39-45).

Q: Lines 54-55 This sentence is not correct “it has been only recently employed, as routine, to improve genetic traits such as milk production, meat, wool and the ability to compete in the Middle East [2] [3]”. The technique is not employed as routine, please change the sentence indicating that the technique is under development for its possible/potential use, on a large scale, for the indicated objectives.

A: We have modified the sentence (lines 56-57).

Q: Line 142 “2.1.2. Embryo transfer”; where is paragraph 2.1.1.?

A: Sorry, the numering was off. We have modified it (lines 238).

Q: Apologies for not being enough detailed. The following information might be helpful so as to provide a quick idea about the context. Authors might report a short summary about the issue of semen collection centers and its critical aspects:

“To better understand provisions of the Reg. 686, it is important to focus attention on some aspects relating to the organization of semen collection centre and points about the biosecurity. Monaco and Lacalandra (2019) [4] provided a brief report on the above-mentioned points, and they have highlighted several drawbacks regarding AI in the dromedary.

The importance of the presence of the responsible veterinarian, as also indicated in the Terrestrial Animal Code [13], and the division into different biosecurity zones (collection room, semen storage, laboratory, etc.) to allow physical separation between the various areas of the centre is emphasised. In addition, the presence of a surveillance plan consistent with the principles of hazard analysis and critical control points (HACCP) and the keeping of all documentation is not negligible. The latter must provide clear evidence that biosafety, surveillance, and traceability practices are effectively applied [4].

Unfortunately, to date, there are no approved and effective freezing/thawing strategies to maintain sperm viability, which is a limitation to the application of artificial insemination on a large scale [14].

It is probably necessary to conduct studies on the screening of infectious diseases and the hygiene of sperm collection and processing [4] [3] [15].

A: Sorry, when making corrections to the first version, some modifications have been lost. We have revised this part (lines 135-175).

Q: Moreover: the author did not address a previous comment about where camel semen could be collected (confined establishment, recognized establishment?) Please clarify the difference in a practical approach base. i.e. for Sheep and Goats semen can also be collected in a confined establishment if certain conditions are assumed (please refer to the related regulation articles): Authors shall report such circumstances and point out that, despite not being considered by the law it is quite difficult that a camel semen collection centre could be built in Europe (cost, animal movements, little benefit due to the low number of Camelidae heads) therefore exception such as the one for rams and bucks shall probably be considered. 

A: The information required has been added (lines 135-154).

Kind Regards

Prof. Michela Pugliese

Reviewer 2 Report

The authors have address my concerns.

Some minor comments to improve the quality of the manuscript are the following

Line 17: check the English, something is wrong in the sentence

Line 59. The management of the bulls is a risk factors for the quality of semen. Please add it and cite Aubè et al 2016, Peer J, and some of the works done by Fatnassi on this point. The poor welfare and the inappropriate management of the bulls is a critical point which should be enlarged, considering the importance of welfare on the health status of the animals and the recent focus of the EU regulation on the welfare

Line 63. Have a look at the recent publications by Skidmonore. Her team is quite successul in embryo transfers

Should also Antibiotic resistance (AMR, and AMR bacteria) be considered within the review? European AHL is focusing a lot on this topic, see recent EFSA opinions

Author Response

Dear Reviewer,

Thank you very much for your time and all your comments.

 We thank you for your precise and thoughtful comments and constructive criticism, which has led to a better manuscript.

We revised the manuscript in relation to the suggestions and more detailed answers are given below.

The changes made in the manuscript to address comments are marked up using the 
“Track Changes” function.

Q: Line 17: check the English, something is wrong in the sentence.

A: The sentence has been modified (lines 16-18).

Line 59. The management of the bulls is a risk factor for the quality of semen. Please add it and cite Aubè et al 2016, Peer J, and some of the works done by Fatnassi on this point. The poor welfare and the inappropriate management of the bulls is a critical point which should be enlarged, considering the importance of welfare on the health status of the animals and the recent focus of the EU regulation on the welfare.

A: We have revised this part based on your suggestions (line 62-69).

Q: Line 63. Have a look at the recent publications by Skidmonore. Her team is quite successul in embryo transfers.

A: We have cited this Author by adding a more recent article. Details are reported in paragraph 2.1.2 (Reference n.o. 20)

Q: Should also Antibiotic resistance (AMR, and AMR bacteria) be considered within the review? European AHL is focusing a lot on this topic, see recent EFSA opinions

A: There are specific regulations on veterinary medicines (Regulation (EU) 2019/6) and medicated feed (Regulation (EU) 2019/4) influencing antimicrobial prescribing and usage throughout EU in food animals in order to fight antimicrobial resistance. Surely, these legislations deserve attention. Antibiotic resistance is an important issue, but in our opinion, we think that it should not be considered in this context. In this article, we focus our attention on Regulation 686.

Kind Regards

Prof. Michela Pugliese

Round 3

Reviewer 1 Report

.